# The Biodiversity Footprint of German Soy-Imports in Brazil

Lukas Mahlich, Christopher Jung and Rüdiger Schaldach *

Center for Environmental Systems Research, University of Kassel, Wilhelmshöher Allee 47,
34109 Kassel, Germany
* Correspondence: schaldac@uni-kassel.de

**Abstract:** By importing agricultural commodities, Germany causes ecological impacts in other countries. One of these impacts is the loss of biodiversity in the producing regions. This paper presents a new method that combines agricultural trade data with land cover and biodiversity data to assess these effects within an agricultural supply chain, in a spatially explicit manner. It considers the current state of biodiversity expressed by the biodiversity intactness index (BII) of the producing region as well as changes over time. As an example, the impacts of German soy imports from Brazil were assessed for the time steps 2004, 2011 and 2018. In the first step, the soybean cultivation area used for exports to Germany was assigned to the respective municipalities by using the TRASE database. In the second step, a GIS-based analysis was conducted to determine the resulting impacts on biodiversity. In 2018, 70% of German soy imports from Brazil originated from municipalities with ecosystems that are not intact anymore (50%) or even severely disturbed (20%). Total German soy imports from Brazil in 2018 reduced the BII by 0.03 percent points compared to 1997. The main advantage of the presented method is that the biodiversity impact is quantifiable for both a commodity and the consuming country.

**Keywords:** bioeconomy; footprint analysis; spatially explicit analysis; biodiversity intactness index; planetary boundaries; supply chain management; land use modelling; land use change; deforestation





## 1. Introduction

Human-induced biodiversity loss threatens valuable ecosystem services and human wellbeing [1,2]. Interventions such as the expansion of agricultural land or settlement areas and the associated habitat changes have led to a massive extinction of species—the so-called sixth mass extinction [2–4]. In this context, land use resulting from agricultural activity is a major driver of global biodiversity loss [5,6]. The impact of agriculture on biodiversity is expected to further increase due to changing consumption patterns and a growing world population [7].

Soybean is the fourth most common arable crop in the world and has experienced a constant and strong expansion in recent decades [8,9], due to the increasing global demand for meat, as soy is mainly used as animal feed [10,11]. Most soy cultivation takes place in Brazil [9], a hotspot for biodiversity [12,13]. By importing soy from that hotspot, Germany contributes to the loss of biodiversity. Germany has a special responsibility in this regard as it has a high ecological impact and is a net importer of (virtual) agricultural land [14,15].

Most supply chains are complex and globalized, with agricultural products often produced far from the point of final consumption, making environmental impacts invisible to consumers and difficult to track [16]. The assessment of these remote effects requires the merging of data on biomass trade with spatial data for determining the environmental impacts in the exporting countries. There are several methods and studies that determine different types of these environmental footprints (see Kastner et al. [17] for a detailed review). For the German bio-economy, Bringezu et al. [18] calculated four footprints related to land, water, forest and greenhouse gas emissions. Studies by Green et al. [19] and

Wilting et al. [20] are examples of a quantification of the effects of global trade on biodiversity. In these cases, the impact on biodiversity is indicated by the mean species abundance (MSA) or a "conservation score". There are numerous studies that use the biodiversity intactness index (BII) as an indicator to assess biodiversity changes in individual biomes, such as tropical forests, orin order to illustrate the benefits of private protected areas [21,22]. However, to our knowledge, there are no studies that use this indicator in the context of footprint analyses along supply chains [17,23,24]. For cocoa, there is one example that allocates the BII to a commodity, but this is solely studied in relation to the type of cultivation, not considering supply chains [25].

In this paper, we propose a new method for assessing supply chains of agricultural products for their impact on biodiversity and present a first application to soy imports by Germany from Brazil. The aim is to combine spatial data of land use, land-use change and biodiversity with data on biomass flows along supply chains in a reproducible and transparent manner.

## 2. Materials and Methods

### 2.1. Data

For our analysis, we used map-based datasets with information on biodiversity, administrative boundaries, and land use. In order to calculate the biodiversity intactness index (Section 2.2), we used data from the literature to describe the impact of land use on changes in biodiversity. Information on soy production and trade flows was derived from an existing global database. Table A1 provides an overview of the data used in our study.

#### 2.1.1. Map Based-Datasets

The boundaries of the administrative units were taken from the Brazilian Institute of Geography and Statistics [26]. As our analysis is performed at the municipality level, the 5570 Brazilian municipalities were used as vector layers.

The land cover and land use of Brazil was taken from the CCI Land Cover map of the Copernicus program of the European Space Agency [27] with a spatial resolution of $300 \times 300$ m per raster cell.

Spatially explicit data on biodiversity were derived from Jenkins et al. [28], which provide information on species richness of three different taxa on maps with a spatial resolution of $10 \times 10$ km per grid cell [28,29]. The three terrestrial vertebrate groups are birds [30], amphibians and mammals [31]. Each grid cell has a species richness value per taxon, with the highest value being 611 for birds, 132 for amphibians, and 199 for mammals [29].

#### 2.1.2. Impact of Land Use on Biodiversity

Population impact indicates the degree to which a particular land use reduces or increases the biodiversity of a formerly undisturbed ecosystem [32]. It can be considered as a characterization factor [33,34]. For our analysis, we derived these data from the Globio3 model [28]. The land use/land cover types of the CCI Land Cover maps were grouped into eight classes and linked to the according population impact (see Table A2 [32,35,36]). Soy is assigned to the land-use category intensive cropland with a population impact of 0.15, meaning that only 15% of the native species' abundance remains. Only urban areas have a lower value of 0.05.

#### 2.1.3. Global Trade and Supply Chain Data

TRASE—Transparency for Sustainable Economies—is a database that maps the production location of agricultural commodities to the importing countries [37]. Due to the good data validity on soybean cultivation and the long period under consideration, soybean from Brazil serves as TRASE's flagship [38]. Importing countries are assigned to Brazilian municipalities using the SEI-PCS approach, thus mapping a country–commodity combination [38–40]. The dataset "SEI-PCS Brazil soy v2.5.0" was used for the analysis,

with supply chain information on German soy imports from Brazil from 2004 to 2018 [41,42]. In addition to the exact municipality of cultivation, detailed information is provided on soybean production quantity, associated land use, GHG emissions, logistics center, import port, export port and their companies [42,43]. Soy exports include soybeans, soybean cake and soybean oil and are quantified by weight as soybean equivalent [41]. Seed production and post-cultivation processes are not considered in our analysis.

### 2.1.4. Data Preparation

For processing spatial data, we used two geographic information systems (GIS): (1) The open source software QGIS 3.18 and (2) the ArcGIS Pro 2.8 software from ESRI Inc., Redlands, CA, USA. According to the TRASE database, 1746 Brazilian municipalities exported soy to Germany between 2004 and 2018. These 1746 municipalities were selected from the administrative boundary dataset and the area was gridded to a raster map with a grid cell size of 300 × 300 m (according to the resolution of the CCI map). The species richness maps by Jenkins et al. were resampled to the same resolution of 300 × 300 m per grid cell. For each grid cell, information on municipality name, species richness and land use class was determined. In addition, we assigned the respective population impact value of the cell's land-use type. These steps were conducted for the years 1997, 2004, 2011 and 2018, which we use as reporting years and steppingstones for our further analysis (see below). With the selection of these years, we use all available data in TRASE and, in addition are able to define time steps of equal length to make our results better comparable among each other.

### 2.2. Biodiversity Intactness Index

The biodiversity intactness index (BII) is an indicator for determining the overall status of biodiversity and was developed by Scholes and Biggs [44]. The BII is defined as the average number of species of a taxonomically and ecologically broad group of species in an area, relative to their number of species in an intact reference ecosystem [44]. The BII varies between 0 and 1. Values close to 1 indicate that the species number is close to the intact reference ecosystem, while lower values indicate the extinction of species. For our analysis, the BII is calculated according to [44] using the following equation:

$$BII = \frac{(\sum i \; \sum j \; \sum k \; R_{ij} \; A_{jk} \; I_{ijk})}{(\sum i \; \sum j \; \sum k \; R_{ij} \; A_{jk})} \tag{1}$$

In this equation, i represents the taxon, k the land-use class and j the ecosystem. Thus, $R_{ij}$ determines the species diversity of taxon i in ecosystem j, $A_{jk}$ the area of land use k in ecosystem j, and $I_{ijk}$ the population impact (see section above), i.e., the impact on the population of species group i by land use k in ecosystem j [32,44]. For each grid cell, there is a species richness value for $R_{ij}$, a cell size value for $A_{jk}$, and a population impact via land use class value for $I_{ijk}$. In order to perform the footprint calculation described in Section 2.3, the BII for each municipality was first determined according to Equation (1), using the prepared raster data. This calculation was conducted for the years 1997, 2004, 2011 and 2018.

The main reason for choosing BII as a biodiversity indicator is its suitability to assess whether the status of an ecosystem is still acceptable within planetary boundaries. Still,,it is unclear to what extent a global or biome-based threshold exists [21,45,46]. The safe range of planetary boundaries can be defined by a value of 80%, i.e., a BII of 0.8 [47,48]. However, the authors consider uncertainties and assume a wide range of variation—with a BII of 0.8 to 0.3 as a potential threshold [47–51].

### 2.3. Calculation of the Biodiversity Footprint

The biodiversity footprint includes two approaches and was calculated for the three reporting years 2004, 2011 and 2018. For each reporting year, we only considered the municipalities that actually exported soy to Germany in that year.

The first approach is an assessment of the exported amounts of soy in relation to the ecological status of the source municipalities. For this purpose, we categorized the source municipalities according to their BII. Reflecting the planetary boundary concept (see Section 2.2.), we distinguish three categories: "intact ecosystems" (BII > 0.8), "non-intact ecosystems" (0.3 ≤ BII ≤ 0.8) and "severely disturbed ecosystems" (BII < 0.3). For each category, we then sum up the amount of soy exported to Germany.

$$BII_{Germany,i} = \frac{A_{soy,\,i}\,[ha]}{A_{municip,i}\,[ha]} \times \Delta BII_{municip,i} \tag{2}$$

$$BII_{Germany,\,cum} = \sum_{i=1}^{n} BII_{Germany,i} \times \frac{A_{municip,i\,[ha]}}{A_{allmunicip}[ha]} \tag{3}$$

The second approach quantifies the contribution of soy exports to Germany to changes of biodiversity in the reporting year compared to a specified base year. This contribution is expressed as total change of BII and the change of BII per ton of exported soy. We conduct a "flow-based" accounting [52] that attributes changes in biodiversity to the responsible land use in each reporting year (in our case soy production exported to Germany) not considering the actor who was initially responsible for the conversion of natural habitat area. The calculation includes two steps delineated in Equations (2) and (3).

In the first step (Equation (2)), the German impact on BII change is determined for each individual exporting municipality ($\Delta BII_{Germany,\,i}$). For this purpose, the BII change within the municipality i between the reporting year and the base year ($\Delta BII_{municip,\,i}$) was determined as described in Section 2.2. The attribution of the German impact is then calculated by multiplying the BII change with the quotient of the area used for producing soy exported to Germany as derived from the TRASE database ($A_{soy,\,i}$) and the total area of the municipality ($A_{municip,\,i}$). Using this simple allocation rule, we managed to relate the size of soy area used for export to Germany to its contribution to BII loss.

In the second step (Equation (3)), the cumulative BII loss in Brazil attributed to soy exports to Germany ($\Delta BII_{Germany,\,cum}$) is calculated. This is achieved by summing up the impact values of the individual exporting municipalities ($\Delta BII_{Germany,\,i}$) each weighted by the quotient of its total area ($A_{municip,\,i}$) and the total area of all exporting municipalities ($A_{allmunicip}$).

In addition, we calculated the contribution of each metric ton of exported soy to BII loss, simply by dividing the total BII loss (see above) by the amount of total soy exports to Germany in the reporting year. These calculations were conducted for each of the three reporting years in relation to a set of different base years (e.g., 2018–2011, 2018–2004 and 2018–1997).

## 3. Results

### 3.1. Soy Exports to Germany and Related Land Occupation in Brazil

In 2018, China was by far the most important buyer of Brazilian soy with a share of almost 60%. Brazil itself was the second-largest consumer of its domestically grown soy with a share of 15%, followed by the EU, which imported 11% of Brazilian soy in 2018 and has not been the main buyer since 2010. In total, Brazil exports soy to 94 countries. Together with the US, they are responsible for 80% of the world's soybean exports [53]. Brazil's historical soy export is shown in Figure A1.

The amount of total soy exports to Germany decreased from 2.74 Mt in 2004 to 1.75 Mt in 2011 and 1.25 Mt in 2018. In 2004, 267 Brazilian municipalities exported soy to Germany, in 2011 the total was 283, and by 2018 it had decreased to 150. The land occupation for soy cultivation for Brazilian exports to Germany was 656,000 ha in 2004, 540,000 ha in 2011 and 324,000 ha in 2018. Germany's share of Brazilian exports has also declined, from 5% in 2004 to 1% in 2018.

*3.2. BII of the Exporting Municipalities*

In 2004, the median BII of the 267 Brazilian municipalities exporting soy to Germany was 0.4784, with the lowest BII of a municipality being 0.1434 and the highest BII being almost 1. In 2011, the median BII of the 283 exporting municipalities was 0.3913. The BII range of the individual municipalities was from 0.1507 to 0.996. In 2018, the number of exporting municipalities decreased to 150 and the median BII increased to 0.6347. The BII of individual municipalities ranged from 0.1566 to 0.9997.

Figure 1 shows the exporting municipalities of 2004, 2011 and 2018 in a spatially explicit manner. The individual municipalities are presented in color according to the respective BII.

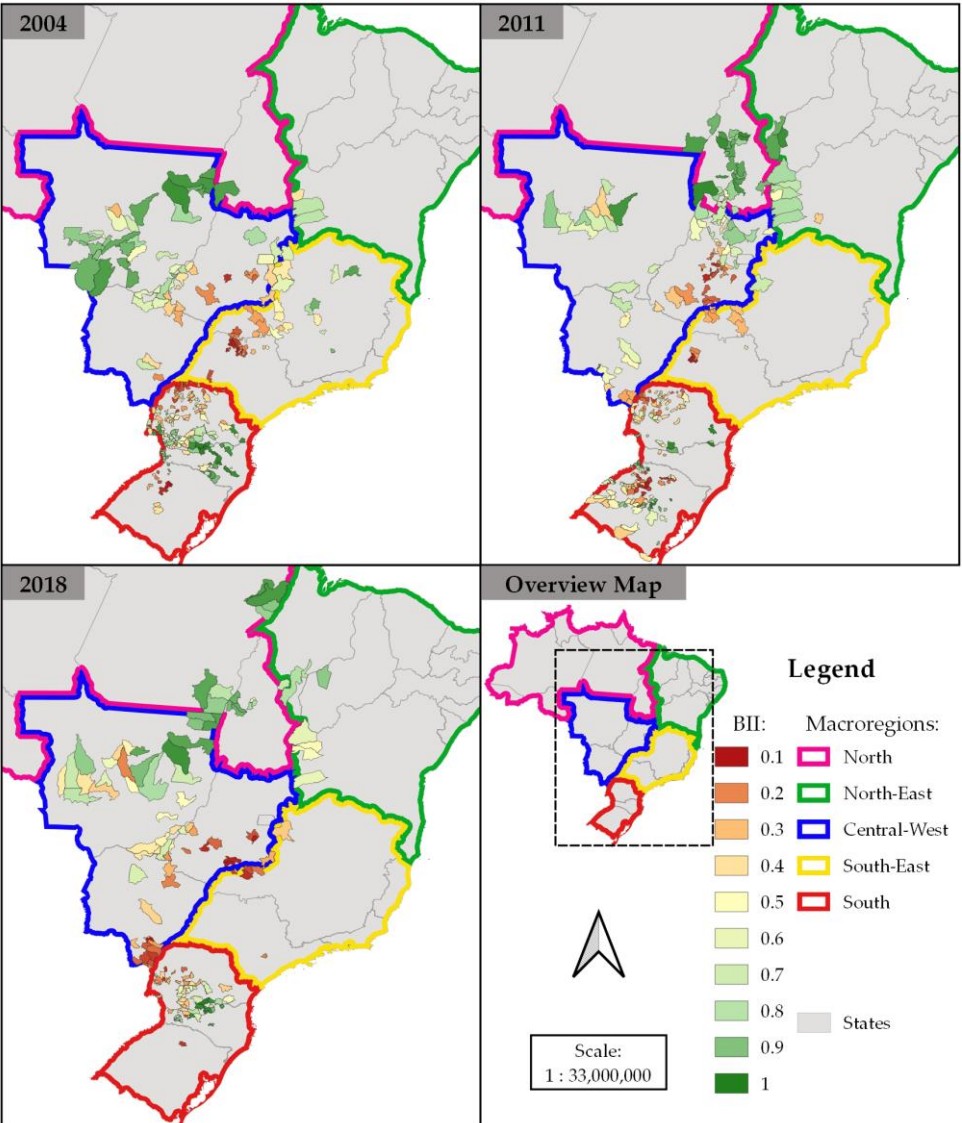

**Figure 1.** BII of Brazilian municipalities that exported soy to Germany in the years 2004, 2011 and 2018.

Figure 2 shows that in 2004 Germany imported 3% from intact, 45% from non-intact and 14% from severely disturbed ecosystems. In addition, 38% originated from unknown areas. This means that TRASE cannot determine the municipalities of soybean cultivation. This value decreases to 1% in 2011 and increases again to 8% in 2018. Imports from intact ecosystems increased both in relative terms, to 11% in 2011 and 22% in 2018, as well as in absolute numbers. The share of non-intact ecosystems was the largest in each year,

reaching 63% in 2011 and 50% in 2018. The share of soy imported from severely disturbed ecosystems increased to 25% in 2011 and decreased to 20% in 2018.

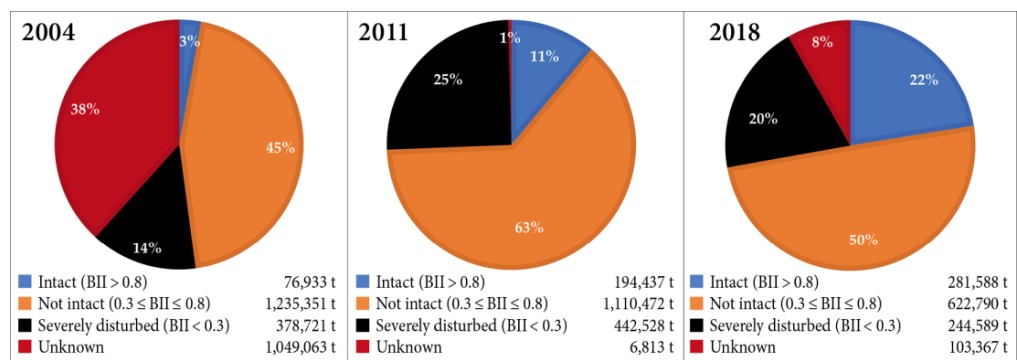

**Figure 2.** Pie charts of the share of intact, not intact, and severely disturbed ecosystems in German soybean imports, measured by species richness. Unknown areas mean that TRASE cannot determine the municipalities of soybean cultivation.

### 3.3. Changes in Brazilian Biodiversity Related to German Soy Imports

BII changes were calculated in seven-year time increments. Since soy cultivation is associated to negative impacts on BII, only BII losses were considered. In this way, parallel land-use changes not associated with soy production could be partially filtered out. For the reporting year, the BII change is calculated in relation to the base year and weighted with exports to Germany, according to the land occupied to produce this amount of soy in relation to the total area of exporting municipalities for that year.

Table 1 shows the impact for the reporting years 2004, 2011 and 2018 of German soy imports on BII changes relative to the respective base years. For example, the value −0.000289 refers to German soybean imports from the 150 municipalities in 2018 and the proportional BII change in these municipalities from 1997 to 2018. The largest decrease in BII occurs in 2011 compared to 1997. German soy imports in 2011 reduced the BII by about 0.05 percentage points compared to 1997. In other words, the 2011 German soy imports, compared to 1997, were responsible for the loss of ~0.05% of the original species richness in the exporting Brazilian municipalities.

**Table 1.** Cross table of the impact of German soybean imports on changes in BII.

| Base Year \ Reporting Year | 2004 | 2011 | 2018 |
|---|---|---|---|
| **1997** | −0.000162369 ΔBII (−):134 (0):10 (+):123 | −0.000458 ΔBII (−):163 (0):7 (+):113 | −0.000289251 ΔBII (−):95 (0):0 (+):55 |
| **2004** | X | −0.0001374 ΔBII (−):156 (0):15 (+):112 | −0.000049182 ΔBII (−):78 (0):0 (+):72 |
| **2011** | X | X | −0.000010129 ΔBII (−):84 (0):1 (+):65 |

(−) indicates how many municipalities with a negative BII are included in the calculation; (0) indicates how many municipalities with an unchanged BII are included in the calculation; (+) indicates how many municipalities with a positive BII are included in the calculation.

The second row of each cell lists the number of municipalities for which the BII of the year under consideration changed negatively, not at all, or positively compared to the base year. The sum of the municipalities in the three classes is the number of exporting municipalities in the year under consideration. A large number of municipalities could enter the calculation with a positive BII change. In 2018, the BII in 78 German soybean import communities has worsened compared to the 2004 base year. Over the same period, the BII has improved in 72 import municipalities. Even if the positive changes are considered,

the magnitude of BII impacts would hardly differ as the positive changes are on such a small scale.

### 3.4. Impact on BII per Ton

The impacts on BII in relation to the amount of exported soy is shown in Table 2. The largest calculated BII change per ton is $-2.621 \times 10^{-10}$ $\Delta$BII/t for 2011 compared to the base year 1997. This is followed by the value of the year 2018 compared to the base year 1997. The smallest effects per ton results for the year 2018 compared to the base year 2011.

**Table 2.** Cross table on BII impact per ton of German soy imports from Brazil.

| Base Year \ Reporting Year | 2004 | 2011 | 2018 |
|---|---|---|---|
| 1997 | $-9.60 \times 10^{-11}$ $\Delta$BII/t | $-2.62 \times 10^{-10}$ $\Delta$BII/t | $-2.52 \times 10^{-10}$ $\Delta$BII/t |
| 2004 | X | $-7.86 \times 10^{-11}$ $\Delta$BII/t | $-4.28 \times 10^{-11}$ $\Delta$BII/t |
| 2011 | X | X | $-8.82 \times 10^{-12}$ $\Delta$BII/t |

## 4. Discussion

### 4.1. Impacts on Biodiversity by German Soy Imports

Our results highlight that in order to develop a comprehensive picture on the effects of German soy imports on biodiversity in Brazil it is important to consider both the state of the ecosystems in a given reporting year as well as changes of biodiversity over time.

Using the BII as an indicator on municipality level allows assessing the individual state of biodiversity related to the planetary boundaries [45] and classifying the soy exports qualitatively. For this purpose, we propose a classification scheme and apply it to track the individual class changes between the reporting years (Figure 2).

In addition, our analyses of BII changes provide information on the actual impact of German soy imports on biodiversity loss and on uncertainties due to certain parameters selected for the calculations. Tables 1 and 2 highlight the importance of three parameters in particular: Base year, export quantity and exporting municipality. For example, the selection of the base year influences the magnitude of the BII changes, while the definition of the total area under consideration influences the overall impact on biodiversity. As we consider the entire exporting area of one year, the selection and number of exporting municipalities also strongly influences the results.

Our calculations of BII changes explicitly do not include the exporting municipalities where increases of BII occur (e.g., due to reforestation efforts). This is for two reasons: (1) Negative effects on biodiversity on the cultivated area persist and cannot be compensated by changes at different locations within the municipality. (2) There is a time lag of positive land use changes on the re-establishment of natural ecosystems and biodiversity [33,54]. In this way, our approach helps to avoid "green-washing" of soy imports. This "green-washing" might occur by accounting for positive effects on biodiversity in exporting municipalities with increasing BII where at the same time, biodiversity is still below the safe limits of the planetary boundaries or where only slight improvements by changes between agricultural land use types took place, e.g., by conversion of degraded pasture to cropland.

An important trend is that between 2004 and 2018, both German and European soy imports and consequently the related impacts on biodiversity are stagnating and even declining to a small extent. German soy imports from intact ecosystems (BII > 0.8) increased while soy imports from non-intact ecosystems ($0.3 \leq$ BII $\leq 0.8$) decreased accordingly. When looking at the import areas, it is noticeable that there is a strong change of cultivation areas between the years. Since soy origin can be easily substituted, it also explains fluctuations in the BII classifications, such as in those severely disturbed.

Nevertheless, total soy production in Brazil strongly increased between 2004 and 2018. At first sight, the European influence seems to lose importance, mainly because

other importers—especially China—have increased their soybean purchases [42]. However, processed products from soy, such as meat and biofuels, are not included in our analysis, which might lead to an underestimation of biodiversity impacts [18].

Since relocation of European and German soy imports to other ecologically valuable world regions should be avoided [55], in the long term, we see that only the reduction of total biomass consumption per person, e.g., through a more plant-based diet and the abandonment of biofuels, can effectively mitigate biodiversity loss without just shifting the problem [56].

### 4.2. Comparison of the Results to Other Studies

Global calculations of mean BII range between 0.846 [47] and 0.785 [49]. Calculations for the Brazilian Cerrado indicate a BII of about 0.7 and about 0.85 for the Amazon [57]. Estimates for the state of Mato Grosso in 2010 assume a BII of 0.59–0.68 and of 0.65–0.86% for Pará [32]. Thus, existing studies are in a similar order of magnitude compared to our study results, indicating that our calculation approach and underlying data are plausible. Since the BII was determined at the municipality level in our analysis, the individual municipalities showed significant differences compared to average values on state level. In addition, the selection of municipalities from which soy is imported isolates the municipality where higher land-use change occurs due to agricultural activities. Therefore, individual municipality BII values may be significantly lower than in previous studies.

An available analysis based on a single crop (cocoa) also using the BII cannot be compared with our results, as it does not look at the supply chain. In a wider spatial scope, Palma et al. [42] analyzed BII changes for tropical and subtropical forests [21]. However, we are not aware of any calculation on BII change that is directly oriented to supply chains and with which we can directly compare our results. Other studies that look at individual crops only refer to land-use change and do not interpret it in the context of biodiversity [17].

### 4.3. Strengths and Limitations of the Applied Method

We see the BII as a suitable indicator for displaying the change in the original biodiversity due to agricultural land use in a spatially explicit manner. The classification of the BII according to the planetary boundaries (see Section 3.2) provides a differentiated view that we think can serve as an important tool for sustainability assessments. For instance, imports from high BII areas can be interpreted as an early indicator of deforestation, or soy imports from municipalities that fall below a specific BII threshold could be avoided in order to discourage neighboring municipalities from destroying their local ecosystems and biodiversity in a similar way.

The quantification of BII changes in combination with biomass trade data as presented in this article can improve the assessment of the impact of the production of agricultural commodities on biodiversity in context of global supply chains. With our case study, we illustrate how to assign these impacts to a specific commodity (soy) and to a produced amount of the commodity. Although the absolute magnitude of the calculated BII impact should be interpreted with caution, the overall spatial pattern could be mapped in a plausible way. Based on that information, it is now possible, e.g., for supply chain managers, to compare agricultural producer regions and to identify exporting regions with preferably low impacts.

Nevertheless, there are several limitations to our study. First, our analysis of biodiversity impacts relies only on relatively rough information on species abundance. Further research should include detailed information, e.g., on endemic species and the role of protected areas. Second, our analysis primarily concentrates on transformation processes in space and time but neglects many aspects of potential impacts of land occupation [49,53]. In this context, future studies should address agricultural management techniques such as slash and burn or double-cropping as well as information on management intensity, e.g., in terms of fertilizer and pesticide application in a more detailed manner. Third, in this paper we focus only on impacts on biodiversity. In order to cover other important

dimensions of sustainability [58], the interlinkages with factors such as indigenous land rights, potential conflicts between biomass exports and local food security as well as other environmental effects of agriculture (e.g., greenhouse gas emissions, water pollution) need to be taken into account to inform decisions in supply chain management. For example, Jarrett et al. [57] highlight that land grabbing of an indigenous area is often accompanied by negative impacts for many species [59].

### 4.4. Application Possibilities

Our macroeconomic approach allows for estimating the impacts of importing countries on biodiversity in exporting countries on a subnational level. In principle, this method can be adapted for any country and commodity. We considered the lack of availability of trade data with a sufficient level of detail as the main limitation of such a spatially explicit analysis.

This method can help in subsequent studies to map impacts of products consumed in Germany or import volumes on Brazilian biodiversity. The calculated values provide characterization factor of BII change per ton of exported soy for the time-periods presented. An application in ecological footprint accounting as part of sustainability monitoring is conceivable [18]. In addition, use within life cycle analysis (LCA) is possible. Here, the effect of land use and land-use change on biodiversity can be investigated via spatially explicit analyses and subsequently made accessible for LCA [60–62]. Such an analysis can support the implementation of conscious supply chain management with systematic selection of exporting municipalities and thus serve as an additional lever for the conservation of biodiversity. In this context, different effects of policies can be examined and presented [63]. Still, from this rather aggregated analysis it is difficult to derive concrete recommendations on potential benefits or drawbacks of further changes of the intensity of soy cultivation, which is why at this point we refer to the existing discussion of land sharing and land sparing [64–66]. However, recent publications highlight that areas with high biodiversity require specific efforts for effective protection [67].

### 4.5. Research Orientation

The presented approach can be used as an estimate of the effect of land-use change on biodiversity with a manageable time investment [33,34]. More detailed remote sensing data and information on actual locations of soy cultivation is desirable. The data of population impact from land management should be refined and made more local [35]. For example, biome or ecoregion related factors could be used [68] to facilitate more elaborated calculations of the BII to derive more detailed information on the population impact [57,69–71]. In addition, it would be desirable to verify ("ground truth") targeted BII calculations with field research.

As mentioned previously, processed products from soy, such as meat and biofuels, should be included in future BII calculations. So far, such products can already be successfully allocated and impacts calculated at country level via MRIO [18]. Extending such calculations to BII at higher spatial resolution is a major challenge for future research.

**Author Contributions:** Conceptualization, L.M. and R.S.; methodology, L.M., C.J. and R.S.; software, L.M.; validation, L.M., C.J. and R.S.; formal analysis, L.M.; investigation, L.M.; data curation, L.M.; writing—original draft preparation, L.M.; writing—review and editing, C.J. and R.S.; visualization, L.M.; supervision, R.S.; All authors have read and agreed to the published version of the manuscript.

**Funding:** This research received no external funding.

**Institutional Review Board Statement:** Not applicable.

**Informed Consent Statement:** Not applicable.

**Data Availability Statement:** Please download the primary data used from others according to the source information. All data processed by us can be found under the following link: https://nc.ufz.de/s/xFNC89sCxaaSpMk (accessed on 26 August 2022). Uses the password: *BII-DEU-BRA*. In case of problems with the download, the data are also available on request from the corresponding author.

**Conflicts of Interest:** The authors declare no conflict of interest.

## Appendix A

**Table A1.** Presentation of all data used with a short description and reference to the source.

| Input Data | Description | Source |
|---|---|---|
| Administrative boundaries | Vector layer of 5570 Brazilian municipalities | [26] |
| Land cover | CCI Land Cover raster layer with a spatial resolution of 300 × 300 m per raster cell | [27] |
| Species richness | Three Raster layers (one for each taxon) with a spatial resolution of 10 × 10 km | [28–31] |
| Population impact | Literature data | [32,35,36] |
| Soy cultivation | Data from TRASE—SEI-PCS Brazil soy v2.5.0 dataset | [41,42] |

**Table A2.** Assignment of CCI Land Cover land-use classes to their individual population impact values derived from [27,32,35,36].

| Land-Use Type | Adapted Land Use Class | CCI-Land Cover—Land Use Class | Population Impact |
|---|---|---|---|
| Cropland | 1 | 10, 11, 12, 20 | 0.15 |
| Pasture (intensive) | 2 | 130 | 0.3 |
| Mosaic agricultural Area/rainforest | 3 | 30, 40 | 0.83 |
| Rainforest | 4 | 50,60,61,62,70,71,72,80,81,82,90,160,170 | 1 |
| Grassland, savannah, shrubland, wetland | 5 | 100,110,120,121,122,140,150,151,152,153,180 | 0.94 |
| Fallow land | 6 | 200,201,202 | 0.5 |
| Urban | 7 | 190 | 0.05 |
| Water | 8 | 210 | 1 |

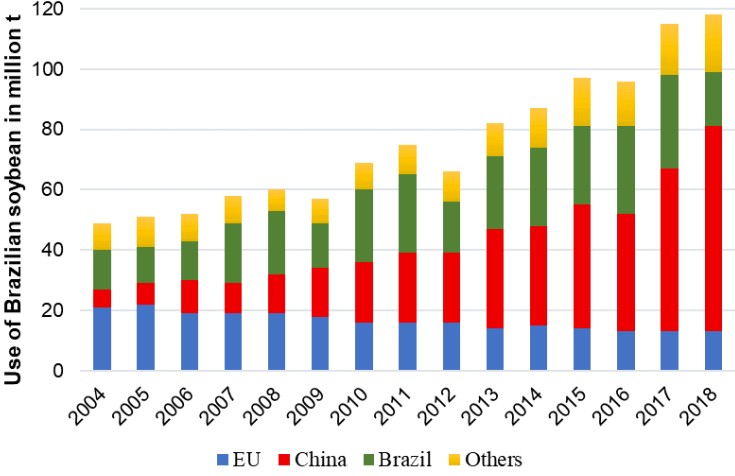

**Figure A1.** Consumption of produced soy in Brazil over time from 2004–2018. The data are used from TRASE [41,42].

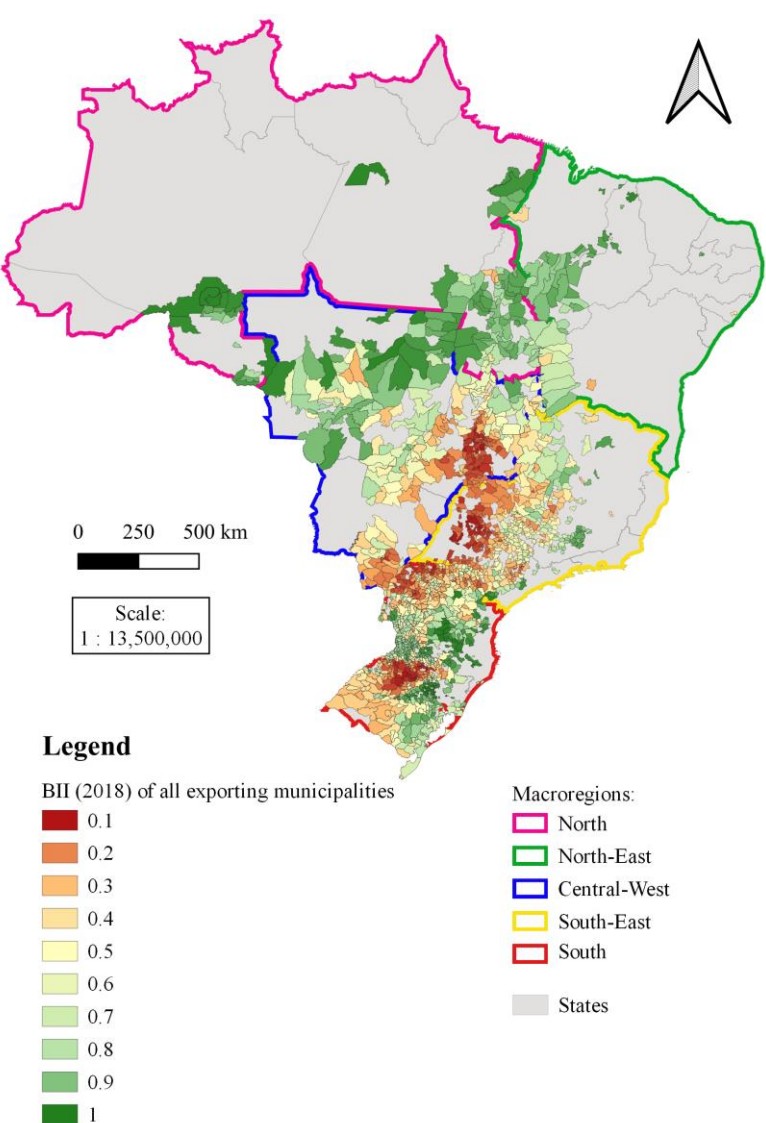

**Figure A2.** Map of Brazil with BII 2018 at the municipality level. All municipalities exporting soy to Germany from 2004–2018 are shown. The BII values are shown in the color gradient.

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
