# Peer review of "The Biodiversity Footprint of German Soy-Imports in Brazil"

_sustainability, doi:10.3390/su142316272_

Round 1

Reviewer 1 Report

Dear Authors,

I would like to thank the author team for their excellent effort on this fascinating research about the Biodiversity Footprint of German Soy-Imports in Brazil.

This paper presents a new method that combines agricultural trade data with land cover and biodiversity data to assess these effects within an agricultural supply chain, in a spatially explicit manner using biodiversity expressed by the Biodiversity Intactness Index (BII) of the producing region as well as changes over time. The research also provides an example of the impacts of German soy imports from Brazil for the time steps 2004, 2011 and 2018. The TRASE database and a GIS-based analysis were used and BII changes were calculated in total and per ton of exported soy. The results presented in this paper are informative and interesting.

A few things need to be clarified.

1.    Abstract should include more results and discussion.

2.    Introduction session:

-        More literature reviews on the Biodiversity Intactness Index (BII) applied in other regions and for other agricultural data other than “soy” and from “Brazil”, …. are needed.

-        Session 2.1: data: Please list all data for the calculation and analysis.

-        Session 2.2. Biodiversity Intactness Index.

-        Please provide details of how to determine/calculate each component of BII, such as Rij, Ajk and Iijk. Please noted that just using citation such as [27,39] is not easy to understand for the readers.

-        Please clarify why the years 1997, 2004, 2011 and 2018 was chosen for analysis in this study?

-        Figure 1: Details of each component/step in the procedure are needed (such as data, sources, … please using Tables or Graphs to present the input data).

-        Statement: “We assume that the contribution of soy exports to Germany to BII changes in Brazil equals the percentage of agricultural land occupied in the total land area of the exporting municipalities”: Please clarify this by citation or providing detail data.

-        Figure 3: Please clarify the “unknown” percentage in the year 2004? Please explain the “trend” of “severely disturbed (BII < 0.3) from 2004 to 2022 and 2018? Is there any association/relation between these figures to the economic crisis in the periods 2004-2018?

Thank you.

Reviewer 2 Report

The paper is well written and interesting to read, however I see only a few minor issues that should be resolved before publishing this paper:

1-      Please revise the keywords, do not use both abbreviation and expansion in this section.

2-      Line 103: Please explain “Population impact” a bit more.

3-      Section 3.1. Please illustrate total soy exports of Brazil over time with a figure.

Reviewer 3 Report

The article is fascinating but contains many ambiguities. Some points should be completed.

1. too weak analysis of the literature on how to calculate footprint (l. 40-45). The Authors of the paper assume in advance that the Reviewers are supposed to know the details of its determination. The article should show how it has been performed until now, and clearly state how the existing ways differ from this new one proposed by the researchers;

2. the new way of calculating footprint is very poorly presented, the primary relationships and novelties are not shown, and what this new method can bring to science is not shown;

3. the benefits of proposing a new footprint calculation methodology are not identified;

4. how and in what categories can the new method be compared to the results obtained by other footprint calculation methods? l.276-288;

5. by inserting other data into the footprint calculation model, can we compare to figures obtained by other methods?

6. if the new method is so innovative, what does it give us if the results are comparable to other methods?

7. no literature items 66

Round 2

Reviewer 3 Report

The authors took into account the reviewers' comments to a sufficient extent. The article is readable, clear, and attractive. I recommend it for publication.